# Stimulation of the left dorsolateral prefrontal cortex with slow rTMS enhances verbal memory formation

**Mircea van der Plas**[1,2☯], **Verena Braun**[1,3☯], **Benjamin Johannes Stauch**[1,4], **Simon Hanslmayr**[1,2]*

**1** School of Psychology, University of Birmingham, Birmingham, United Kingdom, **2** Centre for Cognitive Neuroimaging, Institute for Neuroscience and Psychology, University of Glasgow, Glasgow, United Kingdom, **3** Nuffield Department of Clinical Neurosciences, University of Oxford, Oxford, United Kingdom, **4** Ernst Strüngmann Institute (ESI) for Neuroscience in Cooperation with Max Planck Society, Frankfurt, Germany

☯ These authors contributed equally to this work.
* s.hanslmayr@glasgow.ac.uk

**Data Availability Statement:** All raw data and scripts are available on the open science framework platform: https://osf.io/dyxjv/.

## Abstract

Encoding of episodic memories relies on stimulus-specific information processing and involves the left prefrontal cortex. We here present an incidental finding from a simultaneous EEG-TMS experiment as well as a replication of this unexpected effect. Our results reveal that stimulating the left dorsolateral prefrontal cortex (DLPFC) with slow repetitive transcranial magnetic stimulation (rTMS) leads to enhanced word memory performance. A total of 40 healthy human participants engaged in a list learning paradigm. Half of the participants ($N = 20$) received 1 Hz rTMS to the left DLPFC, while the other half ($N = 20$) received 1 Hz rTMS to the vertex and served as a control group. Participants receiving left DLPFC stimulation demonstrated enhanced memory performance compared to the control group. This effect was replicated in a within-subjects experiment where 24 participants received 1 Hz rTMS to the left DLPFC and vertex. In this second experiment, DLPFC stimulation also induced better memory performance compared to vertex stimulation. In addition to these behavioural effects, we found that 1 Hz rTMS to DLPFC induced stronger beta power modulation in posterior areas, a state that is known to be beneficial for memory encoding. Further analysis indicated that beta modulations did not have an oscillatory origin. Instead, the observed beta modulations were a result of a spectral tilt, suggesting inhibition of these parietal regions. These results show that applying 1 Hz rTMS to DLPFC, an area involved in episodic memory formation, improves memory performance via modulating neural activity in parietal regions.

## Introduction

We are able to encode and store episodes that are rich in detail, filled with information, and highly associative [1]. The first crucial step in forming episodic memories consists of processing the information at hand [2]. Before an event can be stored for later access, it has to be represented [3]. This involves posterior neocortical areas processing different sensory inputs

**Funding:** S.H. was supported by grants from the European Research Council (Nr. 647954), and the Economic and Social Research Council (ES/R010072/1). The funders had no role in study design, data collection and analysis, decision to publish, or preparation of the manuscript.

**Competing interests:** The authors have declared that no competing interests exist.

**Abbreviations:** DLPFC, dorsolateral prefrontal cortex; EEG, electroencephalography; fMRI, functional magnetic resonance imaging; rTMS, repetitive transcranial magnetic stimulation.

under top-down control of prefrontal regions [4,5]. Being able to enhance this process via brain stimulation could prove invaluable not only for therapeutic interventions but also for gaining knowledge about how our brain accomplishes the complex task of forming episodic memories.

The left dorsolateral prefrontal cortex (DLPFC) has been demonstrated to play a role in memory formation (for a review, see [6]). Stimulation at the DLPFC during encoding has been shown to reduce performance on verbal episodic memory tasks [7,8]. These reductions in performance have been mainly achieved with facilitative stimulation protocols (20 Hz stimulation). Thus, it seems that left DLPFC activity might have an inverse relationship to memory performance. Thereby, by inhibiting the left DLPFC, one would expect to see an increase in memory performance. Slow repetitive transcranial magnetic stimulation (rTMS) has been shown to have an inhibitory effect on cortical areas [9–12].

Monitoring the ongoing electrophysiological activity, with electroencephalography (EEG) can inform the mechanisms that lead to a given behavioural observation. We were particularly interested in monitoring the ongoing spectral profile, oscillations in the alpha-beta frequency band typically show a reduction in power during successful memory processing (see [13] for a review), which might reflect more efficient stimulus processing [3].

We here report an incidental finding from the dataset of an existing study [14] in which the authors examined the role of the left DLPFC in voluntary forgetting. We reanalysed their rTMS-EEG dataset and found that 1 Hz rTMS applied to the left DLPFC during encoding of verbal material enhances memory performance. We further found that this rTMS-induced enhancement of memory performance co-occurred with stronger beta-power decreases, a state that is known to be beneficial for stimulus processing [15]. To ensure that the memory enhancing effects of rTMS are replicable, we conducted a second experiment that confirmed the memory enhancing effect of left DLPFC stimulation (experiment 2).

## Results

### Experiment 1: Behaviour

Participants were presented with 2 lists of 10 words per encoding-retrieval run over the course of 12 runs. Following the 6 analysed lists, they were instructed to remember (i.e., keep in mind) the list just presented. After undertaking a short distractor task, participants were asked to recall all words from the 2 word lists just presented. The experimental group received 1 Hz rTMS to the left DLPFC during encoding of the second list, and the control group received stimulation to the vertex (see Fig 1). It is important to note here that the material analysed in this study only represents half of the completed trials by any given participant, as the original study also included lists that were to be forgotten as part of the original paradigm. Trials in these conditions are not further analysed in the context of this study.

**Behaviour.** To test the effect of rTMS on memory performance, we conducted a 2 (List 1 versus List 2) × 2 (DLPFC versus vertex) mixed ANOVA. There was a significant positive effect of DLPFC stimulation on memory performance (main effect rTMS, $F(1,38) = 5.096$, $p = 0.03$, $\eta^2_p = 0.118$) and a significant difference between memory for the first and second lists (main effect list, $F(1,38) = 17.242$, $p < 0.001$, $\eta^2_p = 0.312$). We also found a significant rTMS × LIST interaction ($F(1,38) = 8.837$, $p = 0.005$, $\eta^2_p = 0.189$). Post hoc independent samples $t$ tests revealed that the DLPFC group showed better memory performance compared to the vertex group for words presented during rTMS application (List 2, $t(38) = 2.820$, $p = 0.008$, Cohen's d = 0.892; Fig 2D), but not for words presented before rTMS application (List 1, $t(38) = 1.399$, $p = 0.170$, Cohen's d = 0.443; Fig 2B). Hence, the effects were specific to the application of rTMS to the left DLPFC.

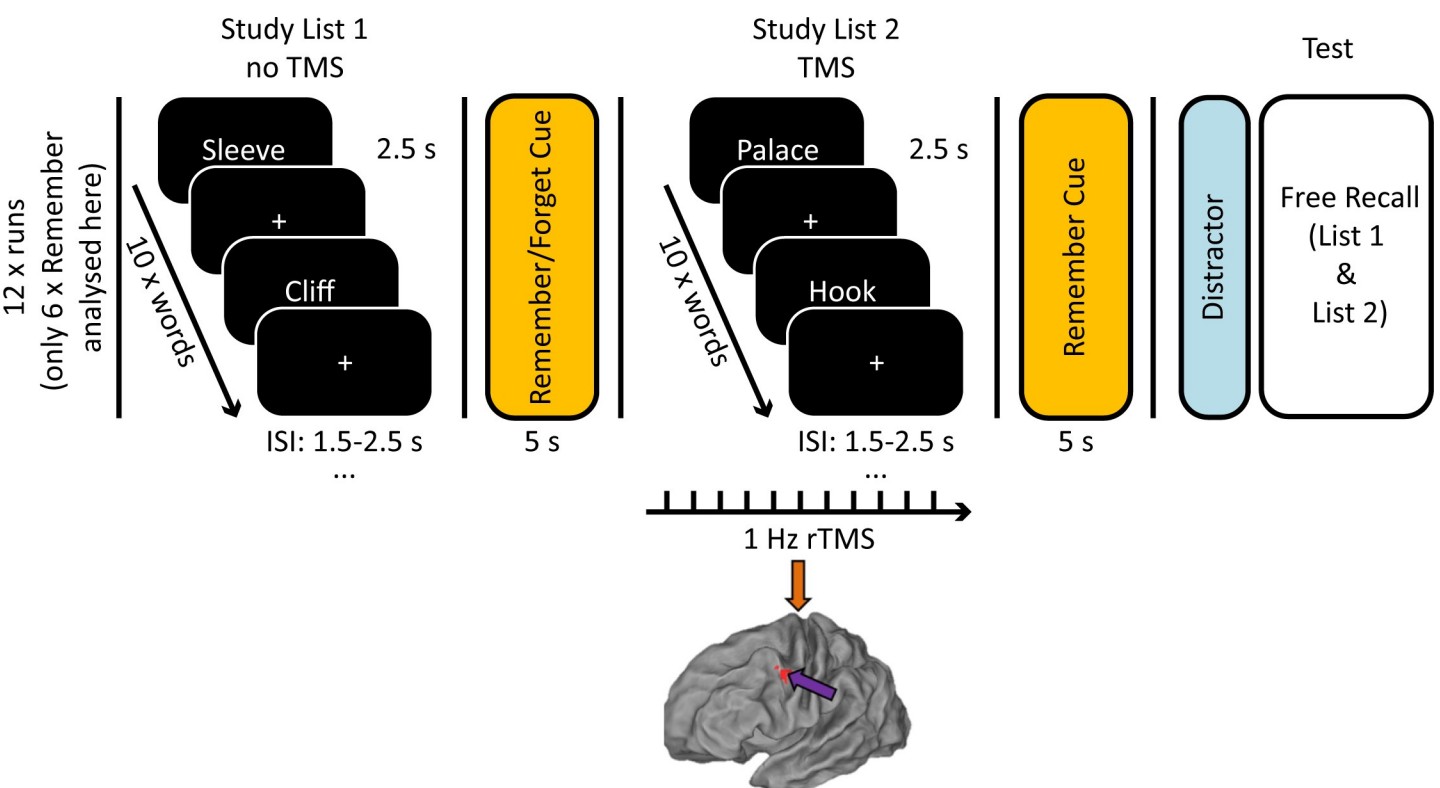

**Fig 1. Experimental design.** Arrows on brain model indicate stimulation site (DLPFC = purple, vertex = orange). Participants were asked to study 2 lists of 10 words over 12 runs. During encoding of List 2, 45 pulses of 1 Hz rTMS were applied to the left DLPFC (MNI coordinates: −45, 6, 39) or vertex. Memory performance was assessed as percentage of correctly recalled words per list. The data and scripts used to generate this figure can be found at https://osf.io/dyxjv/. DLPFC, dorsolateral prefrontal cortex; rTMS, repetitive transcranial magnetic stimulation; TMS, transcranial magnetic stimulation.

In an exploratory follow-up ANOVA, we investigated a possible effect of rTMS on serial position to assess whether left DLPFC stimulation affected the likelihood of recalling a word as a function of its list position [17]. Analysis of serial position curves revealed a significant LIST × POSITION × rTMS interaction ($F_{(9,342)} = 2.435$, $p = 0.011$, $\eta^2_p = 0.06$). To unpack this 3-way ANOVA, we calculated two 2-way ANOVAs for each list separately. These ANO-VAs showed a significant POSITION × rTMS interaction for List 1 ($F_{(9,342)} = 2.703$, $p = 0.005$, $\eta^2_p = 0.066$), but no significant POSITION × rTMS interaction for List 2 ($F_{(9,342)} = 0.893$, $p = 0.532$, $\eta^2_p = 0.023$; Fig 2C). The significant interaction in List 1 was due to enhanced recall rates for late position words in the DLPFC group compared to the vertex group (see Fig 2A). These results suggest that online rTMS to the left DLPFC equally increased memory performance in List 2 regardless of position, whereas for List 1, only late position words benefitted from stimulation.

## Experiment 1: EEG

Poststimulus beta power decreases have repeatedly been associated with successful memory formation [13,18,19]. Therefore, we first tested whether the DLPFC group would show stronger poststimulus (0 to 1 s) beta power decreases (13 to 30 Hz) for words that were later remembered (hits) compared to the vertex group for List 2 trials. In order to test for a difference in this time and frequency window of interest, the data were subjected to a cluster-based permutation test [20]. The results show significantly stronger beta power decreases (13 to 30 Hz)

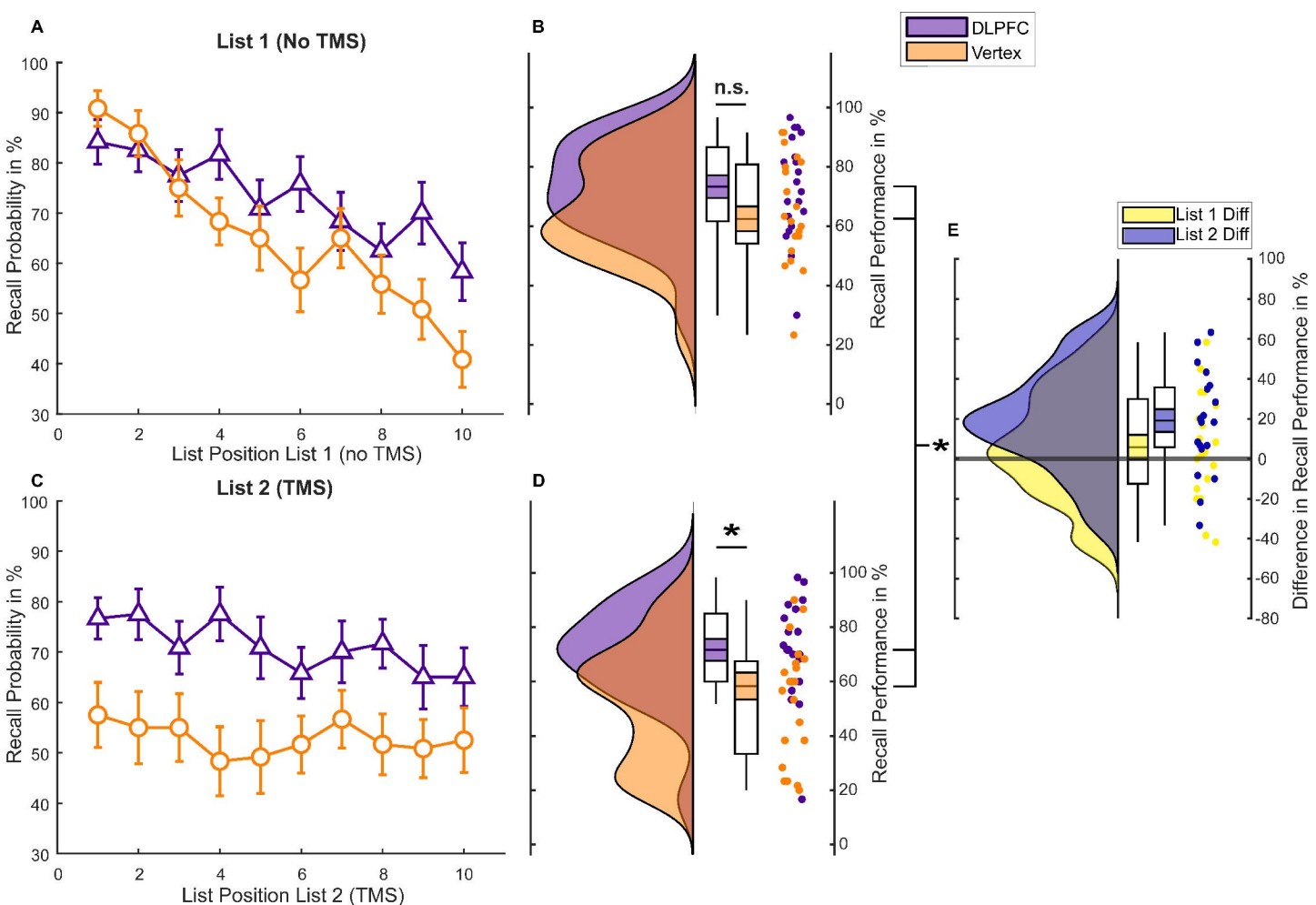

**Fig 2. Behavioural memory performance in experiment 1.** (A) Serial position curve for List 1 words. Error bars depict standard errors of the mean. (B) Raincloud plots of average memory performance for List 1 words across all blocks with paired boxplots [16]. Coloured area within the box plots indicate the standard error, while the circles depict individual data points. (C) Serial position curve for List 2 words. Error bars depict standard errors of the mean. (D) Memory performance for List 2 words. (E) Difference in average memory performance between the DLPFC and vertex condition for each list (List 2 = Stimulation). The data and scripts used to generate this figure can be found at https://osf.io/dyxjv/. DLPFC, dorsolateral prefrontal cortex; n.s., not significant; TMS, transcranial magnetic stimulation.

poststimulus during DLPFC stimulation compared to vertex stimulation. This effect was evident over bilateral posterior sites poststimulus ($p_{corr} < 0.05$, Fig 3B; right poststimulus topography). No effects were obtained for alpha (8 to 12Hz) or theta (4 to 7Hz) frequency bands in this time window. The time frequency plot at this negative electrode cluster, as well as the time course of beta power, is shown in Fig 3A and 3C (for the individual time frequency plots for the DLPFC and vertex condition, see S1 Fig). Beta power showed a clear modulation due to rTMS with regard to word onset in the posterior electrode cluster. Specifically, stronger beta power prestimulus and lower beta power poststimulus were observed during DLPFC stimulation compared to vertex stimulation.

We further explored this beta power modulation to investigate whether it was specific to stimulation trials. Data from −1 s to 1.95 s relative to stimulus onset were split into 6 nonoverlapping time bins (see Fig 3D) for List 1 and List 2 trials for the DLPFC and vertex group, respectively. Data averaged over the significant negative electrode cluster were then subjected to a TIME (time bins) × LIST (List 1 versus List 2) × GROUP (DLPFC versus vertex) ANOVA,

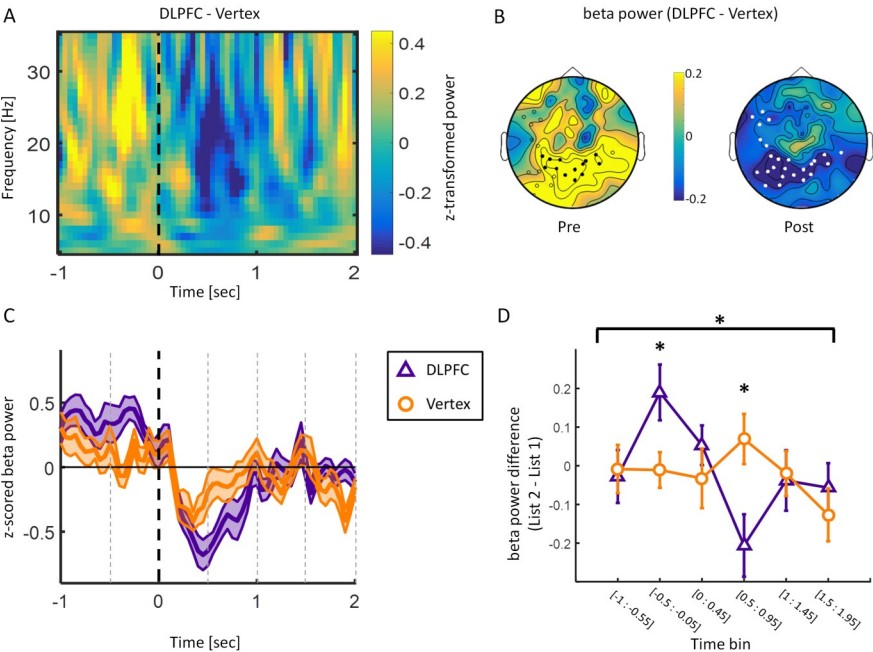

**Fig 3. EEG results (only later remembered trials analysed).** (A) Time frequency plot for the difference between DLPFC and vertex during List 2 encoding averaged over electrode cluster demonstrating a significant negative difference (i.e., less power for DLPFC compared to vertex) between the DLPFC and vertex group in the beta frequency range poststimulus. Dashed line indicates word onset. (B) Topographies depicting beta power (13 to 30 Hz) difference between DLPFC and vertex stimulation in time windows of interest (pre: −0.5 s to −0.05 s; post = 0 to 1 s). White circles depict significant negative electrode cluster poststimulus. Black circles show electrodes within the negative cluster showing a positive difference prestimulus. (C) Time course of beta power (13 to 30 Hz) averaged over the negative electrode cluster shown in B. Shaded area represents standard error of the mean. Black dashed line indicates word onset. Grey dashed lines depict time bins. (D) Beta power difference (List 2 − List 1) over significant negative electrode cluster split by rTMS. Error bars show standard error of the mean. Data were split into 6 nonoverlapping time bins: [−1 s to −0.55 s]; [−0.5 s to −0.05 s]; [0 s to 0.45 s]; [0.5 s to 0.95 s]; [1 s to 1.45 s]; [1.5 s to 1.95 s]. The data and scripts used to generate this figure can be found at https://osf.io/dyxjv/. DLPFC, dorsolateral prefrontal cortex; EEG, electroencephalography; rTMS, repetitive transcranial magnetic stimulation.

which revealed a significant LIST × TIME × GROUP interaction (F(5,190) = 2.707, $p$ = 0.022, $\eta^2_p$ = 0.066). Post hoc independent samples $t$ tests revealed significant increases in beta power prestimulus (−0.5 s to-0.05 s: t(32.347) = 2.384, $p$ = 0.023, Cohen's d = 0.754) and decreases in beta power poststimulus (0.5 s to 0.95 s: t(38) = −2.678, $p$ = 0.011, Cohen's d = −0.847) in the DLPFC group compared to the vertex group (Fig 3D). These results indicate that 1 Hz rTMS at DLPFC modulated beta power predominantly in trials where the stimulation was applied.

## Experiment 1: Spectral tilt versus oscillations

Recent research suggests that some broadband memory-related effects are driven by a change in spectral tilt (i.e., aperiodic components) rather than a change in narrow band oscillations (i.e., periodic components) [21]. To investigate if the above reported effect of DLPFC stimulation on beta power is due to a change in oscillatory activity or a change in spectral tilt, we separated power spectra into periodic and aperiodic components using the FOOOF toolbox (see Fig 4A for schematic representation of the components as labelled by FOOOF) [22]. Moreover, we included components in the alpha band in this analysis, as the raw power spectra exhibited prominent alpha peaks (see Fig 4B).

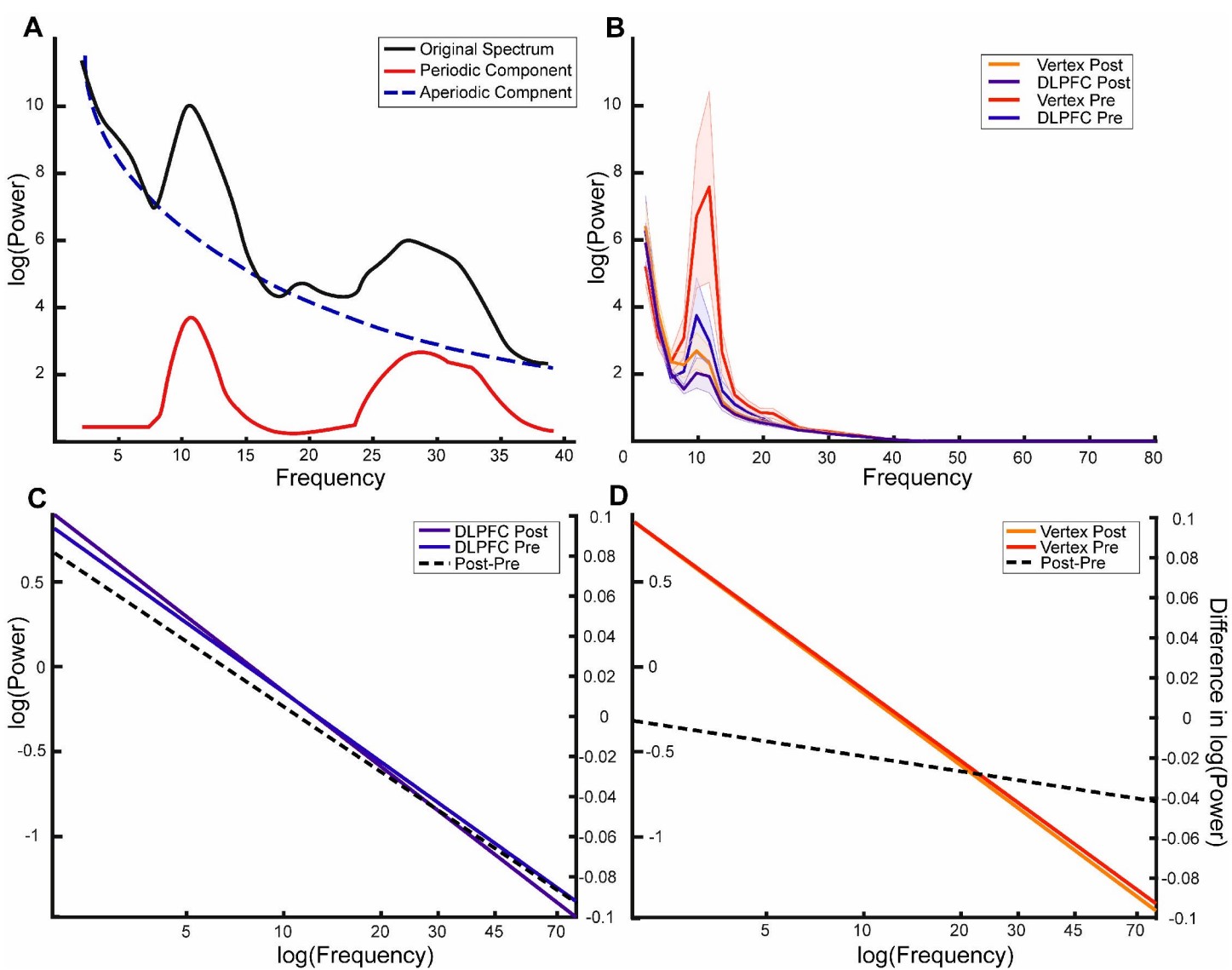

**Fig 4. FOOOF analysis shows that beta effects are nonoscillatory in nature.** (A) Schematic representation of the different components in a given power spectrum. The black line represents a typical power spectrum that is to be separated. The blue line is the corresponding log function following removal of the periodic peaks, thereby representing aperiodic properties of the signal. (B) Power spectra separated by each condition. Shaded area indicates standard error. (C, D) Line plots of the mean aperiodic component before and after item presentation for the DLPFC and vertex condition, respectively. The right axis relates to the plotted post–pre difference (dotted line). The x-axis has been extended for illustrative purposes to highlight the differences in slopes between the difference conditions. The actual fit was performed on data in the 1–40 Hz range. The data and scripts used to generate this figure can be found at https://osf.io/dyxjv/. DLPFC, dorsolateral prefrontal cortex.

We performed a 2 (pre versus post: TIME) × 2 (DLPFC versus vertex: STIMULATION) repeated measurements ANOVA on the periodic and aperiodic components, respectively, with TIME as a within-subjects factor and STIMULATION as a between-subjects factor. We observed a significant interaction effect for the aperiodic component, as reflected by the exponent and offset of the aperiodic component: Exponent: PREPOST × STIMULATION: $F(1,38) = 5.900$, $p = 0.020$, $\eta^2_p = 0.134$; Offset: PREPOST × STIMULATION $F(1,38) = 5.646$, $p = 0.023$, $\eta^2_p = 0.129$ (see Fig 4; for the distributions of the separate components, see S2 Fig). No such interaction effect was observed for the ANOVA investigating the periodic/oscillatory activity in the beta frequency band (PREPOST × STIMULATION: $F(1,27) = 0.652$, $p = 0.426$,

$\eta^2_p$ = 0.024) or alpha frequency band (PREPOST × STIMULATION: F(1,32) = 0.612, $p$ = 0.440, $\eta^2_p$ = 0.019). For both these components, only a TIME effect could be observed (beta: TIME: F(1,27) = 012.267, $p$ = 0.002, $\eta^2_p$ = 0.312) alpha: TIME: F(1,32) = 26.471, $p$ = 0.001, $\eta^2_p$ = 0.453). These results suggest that the interaction observed in the time frequency representation was mainly driven by the aperiodic component, rather than narrow band oscillatory beta or alpha activity. In particular, the results suggest that DLPFC stimulation leads to a steeper aperiodic component where power decreases more quickly as frequency increases.

## Experiment 2: Behavioural replication

Experiment 1 revealed that 1 Hz rTMS to the left DLPFC can increase memory performance for words that were presented during the stimulation compared to a control group. Enhancing long-term memory through rTMS would indeed be an important finding, especially with such a low-frequency stimulation technique that does not require intracranial electrical stimulation or lengthy protocols. Given that our behavioural results were an incidental finding, we attempted an internal replication of the behavioural effect. To rule out any unspecific differences between the groups that might have contributed to the effects, we changed the study design to a within-subjects experiment. Furthermore, in this experiment, the participants as well as the experimenter who interacted with them and scored their memory performance were naïve to the predicted effects of left DLPFC stimulation on memory. Other results of this study have already been reported [23].

To test whether DLPFC stimulation leads to enhanced recall rates compared to vertex stimulation, we conducted a 2 (List 1 versus List 2) × 2 (DLPFC versus vertex) repeated measurements ANOVA. We found a significant main effect for stimulation in the 2 × 2 repeated measurements ANOVA, showing that DLPFC stimulation indeed led to higher memory performance compared to vertex stimulation (main effect rTMS, F(1,22) = 6.778, $p$ = 0.016, $\eta^2_p$ = 0.236). We did not, however, observe a significant effect for list or a significant interaction (main effect List, F(1,22) = 2.943, $p$ = 0.100, $\eta^2_p$ = 0.118; interaction Effect List × rTMS, F(1,22) = 0.009, $p$ = 0.926, $\eta^2_p$ < 0.01). Post hoc $t$ tests revealed a significant difference in recall performance between the DLPFC compared to the vertex condition for List 2 words, during the actual stimulation (t(22) = 2.38, $p$ = 0.026, Cohen's d = 0.496; see Fig 5D). This comparison was not statistically significant for List 1 words (t(22) = 1.754, $p$ = 0.093, Cohen's d = 0.366; see Fig 5B). This pattern suggests that left DLPFC stimulation, once again, led to enhanced memory performance compared to vertex stimulation. Analysis of the serial position curves (Fig 5A and 5C) revealed that recall performance across positions did not differ between the DLPFC and vertex condition in either of the 2 lists (rTMS × LIST × POSITION: F(9,198) = 1.061, $p$ = 0.394, $\eta^2_p$ = 0.046; List 1: rTMS × POSITION F(9,198) = 1.612, $p$ = 0.114, $\eta^2_p$ = 0.068; List 2: F(9,198) = 0.811, $p$ = 0.607, $\eta^2_p$ = 0.036).

For most of the participants ($N$ = 18), the order in which words were recalled was also available. This allowed us to assess the amount of temporal clustering [24] for lists 1 and 2 words and to examine whether DLPFC stimulation affected the amount of contextual error. Such an effect would be predicted by theories implicating the DLPFC in organising memory material into temporal clusters [25]. A 2 (List 1 versus List 2) × 2 (DLPFC versus vertex) repeated measures ANOVA was conducted to determine whether temporal clustering is affected by stimulation. No significant main effects or interaction were observed (main effect for Stimulation: F(1,17) = 0.624, $p$ = 0.440, $\eta^2_p$ = 0.012; main effect for List: F(1,17) = 0.017, $p$ = 0.899, $\eta^2_p$ = 0.003; interaction List × Stimulation: F(1,17) = 0.452, $p$ = 0.511, $\eta^2_p$ = 0.007). To ensure that we did not miss a potential effect of temporal clustering for List 2 items between the

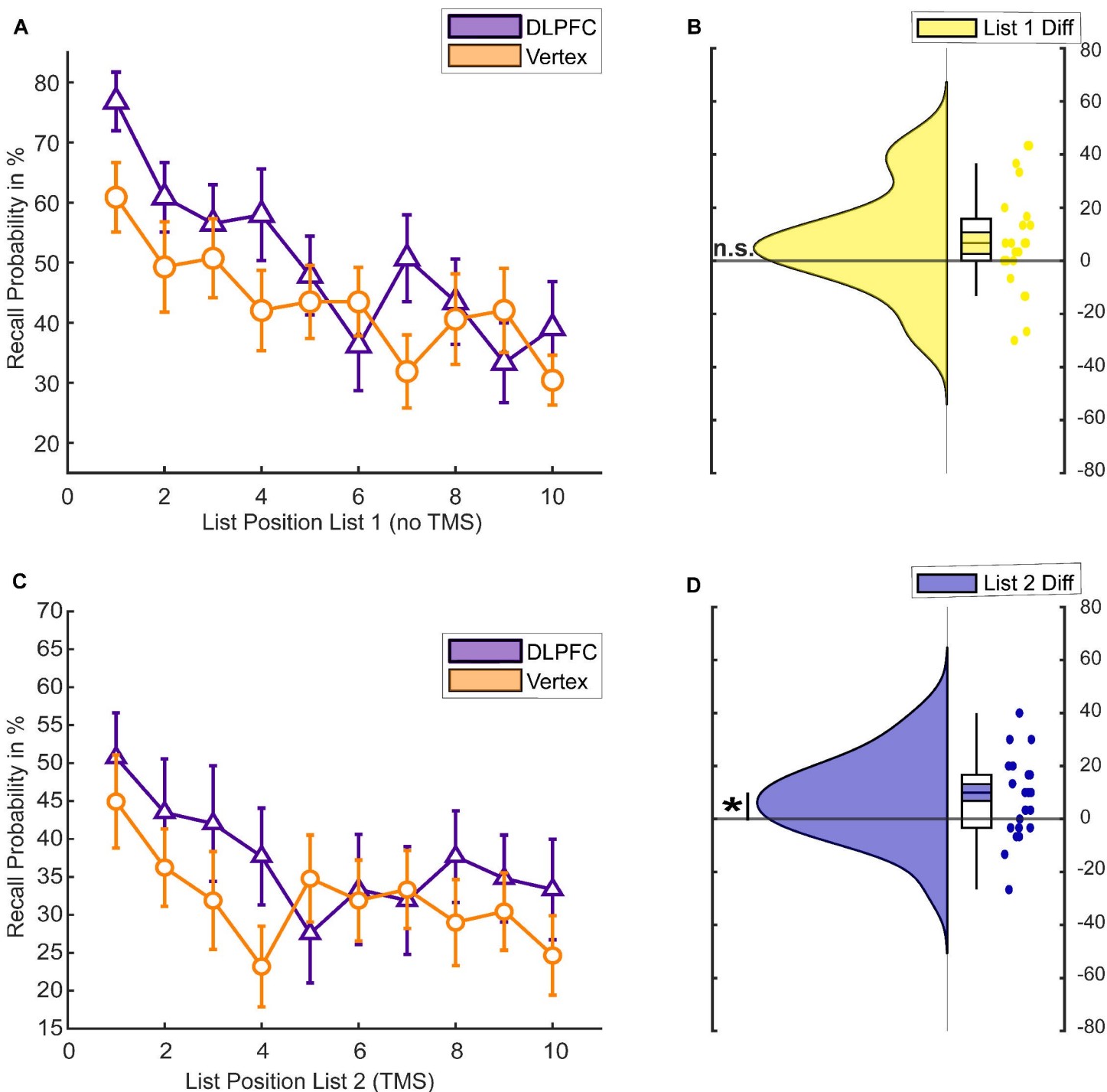

**Fig 5. Behavioural memory performance in experiment 2.** (A) Serial position curve for List 1 ($N$ = 23). (B) Raincloud plots of memory performance for List 1 words (difference between DLPFC and vertex stimulation). Coloured area within the box plots indicate the standard error, while the circles depict individual data points. (C) Serial position curve for List 2. (D) Raincloud plots of memory performance for List 2 words (difference between DLPFC and vertex stimulation). The data and scripts used to generate this figure can be found at https://osf.io/dyxjv/. DLPFC, dorsolateral prefrontal cortex; TMS, transcranial magnetic stimulation.

stimulation conditions, we performed a post hoc follow-up $t$ test on the List 2 only, which also failed to show a significant difference between stimulation conditions (List 2 DLPFC versus List 2 Vertex: t(1,17) = −0.109, 0.914). These results indicate that the memory enhancement

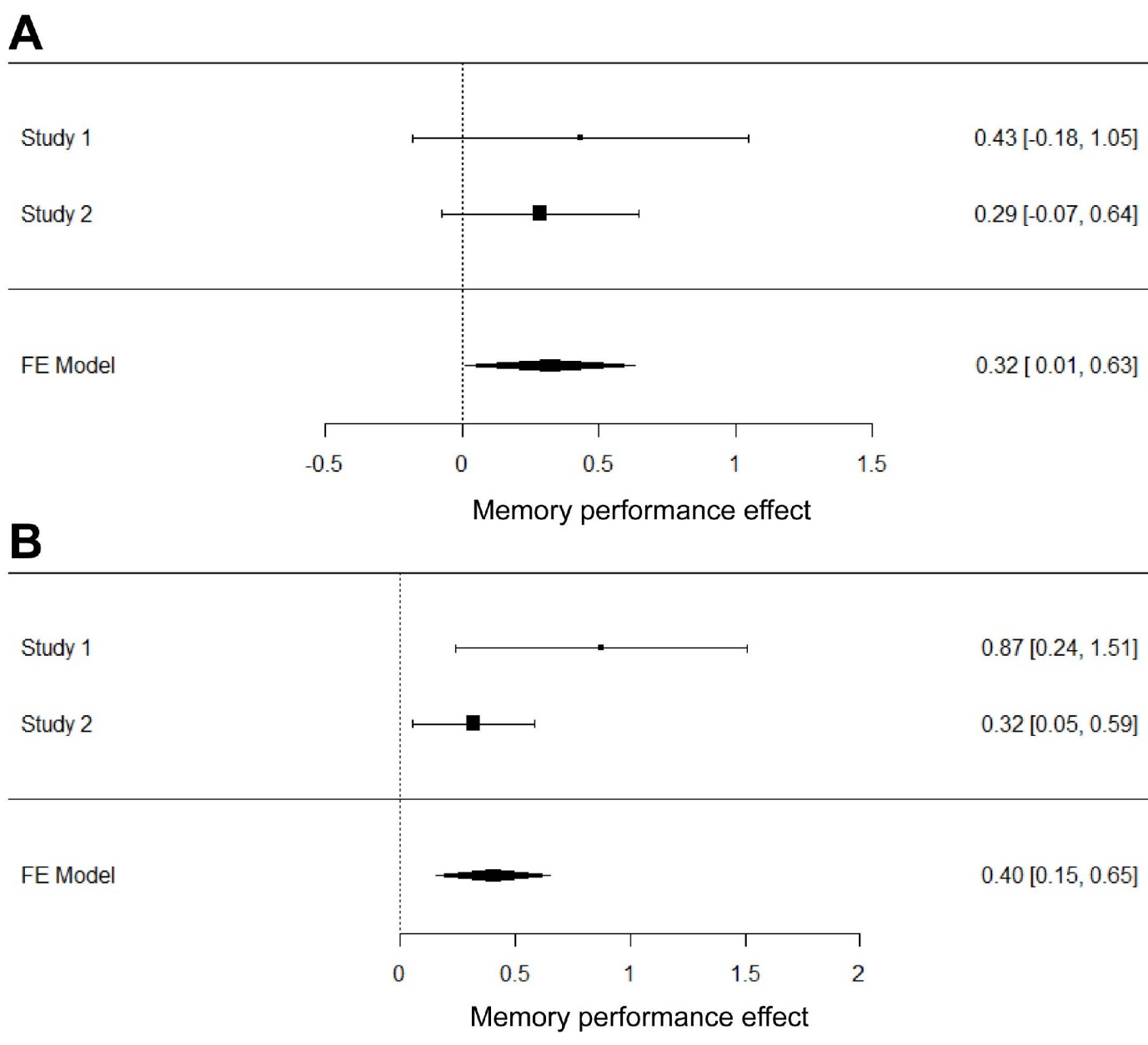

**Fig 6. Meta-analysis of the behavioural results of the first and second experiment.** Forest plots of the meta-analytically combined DLPFC effect for List 1 (A) and List 2 (B) words. Error bars represent 95% confidence intervals; effect sizes were calculated as Hedge's g. The data and scripts used to generate this figure can be found at https://osf.io/dyxjv/. DLPFC, dorsolateral prefrontal cortex; FE, fixed effects.

effect of left DLPFC stimulation cannot be attributed to changes in temporal clustering of the words between or within lists. Rather, DLPFC stimulation seemed to have improved memory performance for each item independently.

Since experiment 1 and experiment 2 used virtually the same paradigm, we performed a continuously cumulative (weighted fixed-effect) meta-analysis over the 2 studies, in order to gain a more accurate estimate of the observed stimulation effect [26,27]. We found that stimulation on the left DLPFC significantly boosts memory performance for both List 1 and List 2 words across the 2 studies (g = 0.32 [0.01, 0.63]; g = 0.40 [0.15, 0.65]) (see Fig 6).

## Discussion

We demonstrated in 2 experiments that 1 Hz rTMS delivered to the left DLPFC during episodic memory encoding boosts memory performance. Participants encoded 2 lists of words and received 1 Hz rTMS during word presentation. In a subsequent free recall test, participants recalled significantly more words from lists in which they received left DLPFC stimulation compared to vertex stimulation. The accompanying serial position and contextual clustering analyses suggest that left DLPFC stimulation enhances stimulus processing at a word-specific level without affecting associations between words. Simultaneously recorded EEG data for the first experiment indicated that 1 Hz rTMS to the left DLPFC strengthened event-related power decreases in the beta frequency band in posterior areas. This was represented by higher beta power before word onset and lower beta power after word onset in the DLPFC group compared to the vertex group. Taken together, our results show that slow rTMS can enhance memory performance and that this memory enhancement effect was associated with increased stimulus-induced beta power decreases, an established correlate of memory function [13].

Power decreases in the alpha/beta frequency range are traditionally associated with stimulus processing in general [28]. While power increases in these frequency bands have been linked to inhibition of irrelevant or potentially interfering information, event-related power decreases (i.e., disinhibition) have been observed over areas actively involved in stimulus processing [29–31]. This beta power reduction has previously been shown to be vital for successful encoding of verbal material [32–34]. This makes sense conceptually, as areas in the MTL can only bind information that has been appropriately processed in downstream neocortical areas [35]. Given its importance in information processing and representation, reduced activity in the alpha/beta frequency bands has been proposed to reflect active involvement of cortical areas during encoding of episodic memories [3,13]. Additionally, TMS has been shown to have network wide effects, which can extend throughout the brain [36,37]. Consequently, it appears that the DLPFC stimulation somehow encourages stimulus processing in parietal and occipital areas, as reflected in the decreased power in those areas. However, a slightly different interpretation could be made considering the result of the analysis separating the periodic and aperiodic components. The observed power changes seem to result from an upward (or clockwise) rotation in the spectral tilt as observed by the increasing exponent and offset components, rather than a change in oscillatory components (see Fig 4). Previous research has suggested that the aperiodic component in electrophysiological signal may be the result of a neural ratio of excitation and inhibition in a local population of neurons [38]. Within this framework, the observed rotation would be associated with increased inhibition [39]. This would imply that the frontal stimulation has an inhibitory effect over the parietal cortex.

This interpretation would be consistent with the fact that we used a stimulation protocol (1 Hz rTMS) that is usually considered to have inhibitory effects on cortical excitability [9,40]. Such an interpretation would be consistent with functional magnetic resonance imaging (fMRI) studies showing that decreased activity in ventral parietal regions is usually positively correlated with memory encoding [41]. This interpretation would also be consistent with other studies reporting a reduction in memory performance when stimulating the left DLPFC with parameters considered to increase excitability (i.e., 20 Hz; [7,8]). The behavioural effects observed in the 2 experiments described here therefore suggest an inhibitory relationship between the left DLPFC and verbal memory encoding. Further, the EEG results suggest that inhibition of the left DLPFC boosts event-related beta power decreases in the service of memory formation. This latter finding suggests that the DLPFC might actively limit the amount of stimulus processing in this memory paradigm. Inhibition of the DLPFC consequently leads to disinhibition in parietal downstream areas. Such reductions in parietal beta power have

previously been associated with an increased capacity of information coded into the neural signal [42]. This increase in potentially coded information would then ultimately result in a better memory performance.

An important caveat of the above interpretation is that it rests on the assumption that online rTMS affects the brain in the same way as offline rTMS does. While rTMS is a method that has been around for decades, most of the mechanistic studies rely on offline effects, where stimulation is first applied and its effects on neural activity or task performance are measured afterwards. This is a consequence of the large artefact a TMS pulse induces in EEG and MRI measurements. Thus, it is conceivable that, while the offline 1 Hz rTMS may have inhibitory after effects, these could result as type of rebound effect from the actual stimulation (and vice versa for the online 20 Hz stimulation employed in the other studies). There has also been a study that have called the inhibitory qualities of 1 Hz rTMS into question [43]. Moreover, the effects of TMS onto the wider network can differ quite drastically from the local effects [36]. Thus, one should not discount the possibility that the parietal decreases might not be a result of modulating the DLPFC activity per se, but rather might result from influencing the memory network as a whole in which the DLPFC plays an important role.

Another possible interpretation that disregards possible facilitative or inhibitory effects of rTMS is that, given our remote effects during left DLPFC stimulation, 1 Hz rTMS may have influenced the functional connectivity between frontal and posterior regions [44]. This enhanced connectivity would then lead to enhanced stimulus processing and improved memory performance as a result thereof. Indeed, a recent study has shown that 1 Hz rTMS can have opposite effects on different networks [45]. Castrillon and colleagues found that while occipital stimulation led to signal propagation to downstream areas, frontal stimulation disrupted network communication. Therefore, extrapolating this finding to the results presented in this paper, it is possible that the parietal beta power decrease is the result of a disrupted network communication, as opposed to local inhibition in the DLPFC per se.

Despite our robust behavioural results, care should be taken when interpreting behavioural rTMS effects. External effects arising from rTMS can influence behavioural measures even when an active control condition is used. DLPFC stimulation, for example, can lead to stronger muscle twitches and distraction than vertex stimulation [46]. This may be experienced as distracting and affect encoding performance accordingly. However, if this was the case, one would expect this to affect performance negatively rather than positively. Furthermore, several studies have found similar effects as those we report here using different stimulation techniques or stimulation in adjacent regions [47–49]. Additionally, Köhler and colleagues [50] showed that when participants received 7 Hz rTMS to the left inferior prefrontal cortex during a semantic encoding task [50], their word memory performance was enhanced. Two control sites were additionally stimulated—the right inferior prefrontal cortex and a right parietal target. Only left prefrontal stimulation resulted in more high-confident hit rates. These findings strengthen our confidence that the results presented are not merely a by-product of unspecific side effects, such as muscle twitches.

Behaviourally, the results in both experiments demonstrate a positive effect of left DLPFC stimulation on memory performance in general. However, the results of the 2 experiments also differed slightly. Considering the first experiment, the memory effect was not only specific to the DLPFC stimulation condition compared to the vertex condition, but also significantly stronger for List 2 words (i.e., those words that were presented during rTMS) as indicated by the significant interaction between words list and stimulation condition. This finding was not replicated in the second experiment where there was no significant interaction between word list and stimulation condition. A possible reason might be carryover effects between lists. However, if this was the case, then the List by Stimulation interaction should also be absent in

the first study. The only difference between the 2 experiments was that Experiment 1 had a between-participant design, while Experiment 2 had a within-participant design. Conceptually, there is no reason why the 2 designs would affect the difference between lists, as carryover effects should still be present when a participant is only exposed to the DLPFC stimulation condition without an accompanying vertex stimulation condition. The results of the meta-analysis do support the possibility that the significant interaction in the first study might be a false positive, because it suggests increases in memory performance for both lists across the 2 studies, thereby suggesting that rTMS during the second list might also enhance memory for previously encoded, but unstimulated items.

Another caveat inherent to the experiment is that due to the lack of a no stimulation condition for List 2, we are unable to completely exclude the possibility that Vertex stimulation reduces memory performance instead of DLPFC enhancing memory performance. A previous study using 1 Hz TMS and measuring fMRI BOLD signal concurrently showed that vertex stimulation does not affect the wider brain other than minor local changes, suggesting that vertex stimulation is a good control site [51].

Lastly, as we analyse data recorded in directed forgetting paradigms, it is unclear if our results generalise to other types of memory tasks. However, considering other work on DLPFC stimulation and episodic memory, the involvement of the DLPFC in episodic memory encoding in general seems to hold across tasks [6–8]. Future research could clarify this by stimulating the DLPFC with 1 Hz rTMS during more general episodic and relational memory tasks.

## Conclusions

Our results indicate that 1 Hz rTMS applied to the left DLPFC during encoding of verbal material can enhance memory performance. This effect was linked to a well-known physiological correlate of memory formation: beta power decreases. Given the need for replication studies in general [52] and for brain stimulation effects in particular [53], we set out to replicate the initial incidental finding. In order to control for interindividual differences [54–56], we replicated our original result in a within-subjects investigation. The results of this second experiment replicated the memory enhancement effect resulting from 1 Hz left DLPFC stimulation. Therefore, online 1 Hz rTMS at left DLPFC appears to be an effective means of enhancing cognitive function in a memory task with potential applicability ranging from basic research to clinical intervention. Future studies should further explore how exactly 1 Hz rTMS to the left DLPFC gives rise to more pronounced beta power decreases in posterior areas and enhanced memory as a result thereof.

## Material and methods

### Experiment 1

**Participants.** The data reported here were collected as part of a larger study (reported in [14] experiment 2). A total of 48 healthy human participants were tested, and participants were randomly assigned to one of the 2 stimulation conditions. After artefact rejection and inspection of the EEG data, 40 participants remained in the sample, resulting in 20 participants per group (DLPFC group: mean age = 21.7, range 18 to 26, 8 males; vertex group: mean age = 22.3, range 18 to 27, 6 males). All participants were right handed, had normal or corrected-to-normal vision, reported no history of neurological disease or brain injury, and were screened for contraindications against rTMS [10]. Written informed consent was acquired from each participant prior to the experiment. The study was approved by the ethics committee of the University of Konstanz (Project ID: "How the synchronized brain forms enduring

memories") and conducted in accordance with the principles expressed in the Declaration of Helsinki.

**Task and stimulus material.** The stimulus material consisted of 240 nouns derived from the MRC Psycholinguistic Database [57]. The material was translated into German and divided into 24 lists of 10 words. The lists were matched according to word frequency, number of letters, number of syllables, concreteness, and imageability [14]. The presentation of the lists was counterbalanced across participants. Each list was presented equally often across 4 conditions (Forget List 1, Forget List 2, Remember List 1, and Remember List 2). The data were collected as part of a study that focussed on the causal involvement of the left DLPFC in voluntary forgetting (reported in [14]; experiment 2). Participants performed 12 encoding-recall runs. In each run, participants were presented with 2 lists of 10 words. After having studied the first 10 words, a cue was presented for 5 s, prompting participants to either forget the previously studied words or to continue remembering this list. The second list of 10 words was always followed by a remember cue. For this study, only the 6 remember runs, i.e., runs in which the first and second lists had to be remembered, are included in the analysis. The words were presented in a randomised order one at a time for 2.5 s, with a variable interstimulus interval of 1.5 to 2.5 s (during which a fixation cross was shown). After a short distractor task of 2 min (counting backwards in steps of 3 from a random number), participants were asked to freely recall as many words from this run as possible in any order. Participants' reponses were recorded manually by the experimenter outside of the EEG room.

**rTMS.** During encoding of List 2, 45 pulses of 1 Hz rTMS were applied at 90% resting motor threshold. One group of participants received rTMS to the left DLPFC, while the control group received rTMS to the vertex. The vertex was chosen as a control site, as it has been shown to not have any wide-ranging network effects for 1 Hz stimulation [51]. The rTMS pulses and stimulus presentation were not synchronised by the experiment. Due to the nature the ISI being randomly chosen as a multiple of 0.25 s, there appeared to be a weak 4 Hz rhythm present (see S3 Fig). However, this bias did not systematically differ between stimulation conditions and therefore cannot explain the observed behavioural effects. rTMS was delivered using a Magstim Rapid2 stimulator with a figure-of-eight air filmed cooled coil (magstim; www.magstim.com). Prior to the main experiment, individual T1-weighted MRI scans were acquired with a 1.5T Philips scanner. In order to assure that the exact regions of interest were targeted, the stimulation was guided by a neuronavigation system (ANT-Visor; www.ant-neuro.com). Individual MRI scans were coregistered with the position of the rTMS coil, and the precise targeting of the stimulation sites was monitored throughout the experiment. The coil was approximately angled 45˚ from the midline axis of the participant's head with the handle pointing backwards and laterally. The MNI coordinates for DLPFC stimulation were x = −45, y = 6, z = 39 [14].

**EEG recording and preprocessing.** EEG was recorded throughout the task from 128 electrodes in an equidistant montage (ANT; www.ant-neuro.com). Participants were seated in a shielded room, and data were recorded with a DC amplifier (ANT) at a sampling rate of 2,048 Hz; data were offline re-referenced to average reference. Individual electrode positions were digitised at the beginning of the experiment (Xsensor, ANT). EEG data were preprocessed and analysed using Fieldtrip [58]. Due to excessive artefacts in the EEG during rTMS [59], List 1 (no rTMS) and List 2 (during rTMS) trials were preprocessed separately. Preprocessing of rTMS-EEG data followed the guidelines and procedure outlined by Herring and colleagues [60] and described on the Fieldtrip tutorial website (https://www.fieldtriptoolbox.org/tutorial/tms-eeg/). EEG data were first cut into segments of −0.9 s to 0.9 s around the rTMS pulses. Data were visually inspected, and data around the rTMS artefacts resulting from ringing and recharging of the stimulator were removed from further analysis, as these can impact the

performance of the subsequent preprocessing steps. The epoched data were subsequently subjected to an independent component analysis (runICA). This allowed the removal of rTMS-related artefacts, eye blink, eye movement, and other remaining artefacts. Any missing data were interpolated with a cubic interpolation algorithm to avoid artificially induced artefacts in the data. The cleaned data epoched around word onset (−2 s to 4 s) were then downsampled to 500 Hz. A low-pass filter (40 Hz cutoff) was applied, and the data were visually inspected for remaining artefacts. Missing and rejected channels were interpolated (mastoids were removed resulting in 126 channels). For trials without rTMS (List 1), data were epoched −2 s to 4 s around the onset of the word, downsampled to 500 Hz, and low-pass filtered (40 Hz cutoff). After visually inspecting the data for artefacts, an ICA was applied in order to identify and remove ocular and muscle artefacts. The cleaned data were again visually inspected.

## Data analysis

**Behavioural analysis.** In order to assess the effect of stimulation on recall performance, a mixed ANOVA with the within-subjects factor LIST (List 1 and List 2) and the between-subjects factor rTMS (DLPFC and vertex) was performed. We further tested whether DLPFC stimulation influenced the likelihood of recalling words as a function on a words' list position. To this end, serial position curves were calculated [17]. For every participant at every list position, we coded whether a word was later recalled (1) or not (0). This was done for all 6 encoding-recall runs and subsequently averaged for every participant over the 6 runs. These data were then subjected to a 2 (DLPFC versus vertex) × 10 (position in list) × 2 (List 1 or List 2) ANOVA.

**EEG analysis.** EEG data (−1.5 s to 3 s) were subjected to a time frequency decomposition (2 to 35 Hz in steps of 1 Hz) using Morlet wavelets (width 7) and z-transformed per trial across time for each participant, within each stimulation condition, to enable analysis of post- as well as prestimulus activity [60]. Since we analysed the data in the context of an increased memory performance, which, according to the sync/desync hypothesis, should be characterised by cortical alpha/beta power decreases, only negative clusters were expected [3,61]. Therefore, data from the DLPFC and vertex group were subjected to a 1-tailed cluster-based permutation test, averaged over beta (13 to 30 Hz) and the poststimulus time window of interest (0 to 1 s). Alpha values were set to 0.05. All further analyses were conducted on the electrode sites identified as showing significant differences in beta between the 2 conditions.

To ensure that any observed effects were specific to stimulation trials, an additional analysis was performed comparing the List 1 and List 2 trials for the DLPFC and vertex groups, respectively, in a time window from −1 s to 1.95 s relative to stimulus onset. This time window was split into 6 nonoverlapping time bins. The data were then analysed using a TIME (time bins) × LIST (List 1 versus List 2) × GROUP (DLPFC versus vertex) ANOVA accompanied by post hoc independent samples *t* tests (see S1 Text for a control analysis regarding potential trial imbalances).

The properties of observed power changes were further investigated using the FOOOF toolbox [22]. This method uses simultaneous fitting of the aperiodic spectrum component as well as spectral peaks. For this, we analysed a 1- to 80-Hz band-pass filtered signal in the time window of interest (resulting from the time frequency analysis) and an identically sized time window before stimulus presentation. This time window was chosen to minimise any effects the filtering process might have on the frequency spectrum. The model was subsequently fit using a frequency range of interest of 1 to 40 Hz to optimise fits for the low-frequency (alpha and beta) bands of interest. These components were then analysed separately. We performed a 2 × 2 mixed repeated measure ANOVA (Pre versus Post (2) word presentation × DLPFC

versus vertex (2) stimulation, for each component (the aperiodic exponent, the offset and the periodic peak power). Additionally, a control analysis was performed to ensure that there were no differences in model fits or residuals between the different FOOOF models that could alternatively explain any of the effects presented in this study (see S1 Table).

## Experiment 2

The data of experiment 2 were part of a larger study that focussed on replicating the effect of rTMS on directed forgetting and are reported elsewhere (see [23]).

**Participants.**  A total of 24 healthy human participants took part in this experiment (mean age = 19.04, range 18 to 28, 5 male). All participants were right handed, had normal or corrected-to-normal vision, reported no history of neurological disease or brain injury, and were screened against contraindications against rTMS [10]. Written informed consent was acquired from each participant prior to the experiment, and participants were fully debriefed at the end. The protocol was approved by the ethics committee of the University of Birmingham (Project ID: ERN_14–0651) and conducted in accordance with the principles expressed in the Declaration of Helsinki.

**Task and stimulus material.**  In this study, the participants as well as the experimenter interacting with the participants were blind towards the hypotheses.

A total of 240 nouns were derived from the MRC Psycholinguistic Database [57] and divided into 24 lists of 10 words. As in experiment 1, the lists were matched according to word frequency, number of letters, number of syllables, concreteness, and imageability [14]. The presentation of the lists was counterbalanced across participants so that each list was used equally often across 8 conditions (DLPFC–Forget List 1, DLPFC–Forget List 2, DLPFC–Remember List 1, DLPFC–Remember List 2, vertex–Forget List 1, vertex–vertex List 2, vertex Remember List 1, and vertex–Remember List 2). Participants performed 12 encoding-recall runs, split by stimulation condition. Whether the 6 DLPFC runs or the 6 vertex runs were conducted first was counterbalanced across participants. The task was the same as in experiment 1. For this study, only the 3 remember runs per stimulation condition are included in the analysis. Participants' responses were recorded manually inside the testing room.

**rTMS.**  The same stimulation parameters were used as in experiment 1. However, in this experiment, participants received both DLPFC and vertex stimulation in a blocked manner. The stimulation was delivered using a Magstim Rapid stimulator with a figure-of-eight coil (magstim; www.magstim.com). Prior to the main experiment, individual T1-weighted MRI scans were acquired using a 3T Philips Achieva MRI scanner. In order to assure precise stimulation, individual MRI scans were coregistered with the position of the rTMS coil, and the stimulation was guided by a neuronavigation system (Brainsight; Rogue Resolutions; https://www.rogue-resolutions.com). The coil was held in place manually, and the precision of the stimulation was monitored throughout the experiment. The same MNI coordinates as in experiment 1 were used.

**Temporal clustering.**  To investigate whether the observed memory effects could be explained due to contextual effects resulting from the stimulation, we calculated temporal clustering scores per participant for each respective list and stimulation condition (procedure is based on [62]). This procedure can be summarised with the formula:

$$\sum_{n=1}^{R} |ObservedDistance_{(n)} - ExpectedDistance|$$

where the observed distance was defined as the absolute difference between the observed recall position and the position during encoding for each subsequently recalled item ($R$). For

example, if a participant recalls an item in the third position and subsequently recalls an item in the fifth position, the observed distance would be 2. The expected difference is the distance value that would be expected during optimal temporal clustering (Expected difference = 1; e.g., one would expect the fourth item to be recalled following the third item yielding a difference of 1).

This yielded a temporal clustering value for each list and condition per participant (see S2 Table for values per condition). The items in List 1 were coded with the numbers 1 to 10, while items belonging to the second list were coded with numbers 11 to 20. These were then directly compared to each other using a 2 × 2 repeated measure ANOVA (List × Stimulation Condition).

**Meta-analysis.** In order to combine the effect of stimulation over the 2 studies, a cumulative meta-analysis of the stimulation effect for the List 1 and List 2 items was performed using the R-package metafor [26]. The analysis was performed by computing effect sizes (Hedge's g) for the individual relevant *t* tests (independent and dependent for study 1 and 2, respectively), which were then used to run a weighted fixed-effect meta-analysis [26,63].

## Supporting information

**S1 Fig. Time frequency representations for List 2 trials during encoding for the DLPFC and vertex stimulation condition, respectively.** The plots contain the averaged activity from selected channels represented by red dots on the accompanying topography. Selected channels are characterised by a significant difference in beta power (i.e., less power in the DLPFC condition compared to the vertex condition). Word onset occurred at 0 s (indicated by dashed line). The data and scripts used to generate this figure can be found at https://osf.io/dyxjv/. DLPFC, dorsolateral prefrontal cortex.
(DOCX)

**S2 Fig. Raincloud plots of the components resulting from the FOOOF analysis for the DLPFC condition (purple) and the vertex condition (orange), for the pre- and poststimulus period, respectively.** Each raincloud plot is paired with its respective box plots. Coloured areas within the box plots indicate the standard error, while the circles depict individual data points for each participant, respectively. The same participants for the pre- and post-time windows are connected by a line. The thick line illustrates the change in mean from pre to post. (A) Raincloud plot of the alpha periodical component. (B) Raincloud plots of the beta periodical component. (C) Raincloud plots per stimulation condition for the offset of the aperiodical component at 0 Hz for pre- and poststimulus period word onset time windows, respectively. Yellow line represents an identical aperiodical component with an increased offset. (D) Raincloud plots per stimulation condition for the Exponent of the aperiodical component Hz poststimulus period windows, respectively. Yellow line represents an example for an identical component with a larger exponent. The data and scripts used to generate this figure can be found at https://osf.io/dyxjv/. DLPFC, dorsolateral prefrontal cortex.
(DOCX)

**S3 Fig.** (A) Raincloud plot of time difference between the first occurrence of a TMS pulse post-word presentation for every trial ($N = 2,400$; 1,200 per condition). Coloured areas within the box plots indicate the standard error, while the circles depict individual data points for each participant, respectively. A slight 4-Hz bias in timing is visible in both conditions based on how the ISI was implemented. With a perfectly random ISI, a uniform distribution would be expected. However, a 2-sample Kolmogorov–Smirnov test confirmed that these 2 distributions do not statistically differ from each other (k-s statistic: 0.0295; $p = 0.6709$). The data and

scripts used to generate this figure can be found at https://osf.io/dyxjv/. DLPFC, dorsolateral prefrontal cortex; TMS, transcranial magnetic stimulation.
(DOCX)

**S1 Text. There was a considerable difference in the number of List 2 hits between the DLPFC and the vertex group because of the enhanced memory performance in the DLPFC group.** (DLPFC: mean = 23.1, SD = 7.48; vertex: mean = 17.25, SD = 8.48). Power is not systematically biased by trial numbers, but we nevertheless tested whether this difference in trial numbers might have contributed to the observed effects. To this end, we randomly selected trials for each participant from the DLPFC group and matched these to the number of trials from participants in the vertex group, ensuring that both groups have exactly the same trial numbers (mean: 17.25, SD: 8.48). As our main comparison of interest was the difference in beta power (13–30 Hz) between the DLPFC and vertex group for List 2 trials, we conducted independent samples $t$ tests for data 0–1 s after word onset averaged over the negative electrode cluster identified earlier. This procedure was repeated 100 times, every time randomly selecting new subsets of trials for the DLPFC group. Approximately 100 $t$ tests on adjusted trial numbers revealed t values ranging from −3.9 to −2.377 (critical t for independent samples $t$ tests = 2.023; df = 38). This analysis demonstrates that the difference in poststimulus beta power decreases for List 2 words was not driven by differences in trial numbers. DLPFC, dorsolateral prefrontal cortex.
(DOCX)

**S1 Table. To ensure that any observed effects of the FOOOF analysis are legitimate and not a result due to differences in model fit, we ran two 2 (within factor time: pre vs post) × 2 (between factor stimulation, DLPFC vs Vertex) mixed ANOVAs as control analyses.** Since none of the factors showed a significant difference, the effects cannot be attributed due to differences in model fits. DLPFC, dorsolateral prefrontal cortex.
(DOCX)

**S2 Table. Mean temporal clustering values per condition, with the accompanying standard deviation.**
(DOCX)

## Acknowledgments

We would like to thank Benjamin Griffiths for advice on temporal clustering analyses and Nora Oehler for help with data collection for experiment 1.

## Author Contributions

**Conceptualization:** Verena Braun, Simon Hanslmayr.

**Data curation:** Verena Braun, Simon Hanslmayr.

**Formal analysis:** Mircea van der Plas.

**Funding acquisition:** Simon Hanslmayr.

**Investigation:** Mircea van der Plas, Verena Braun, Benjamin Johannes Stauch, Simon Hanslmayr.

**Methodology:** Mircea van der Plas, Verena Braun.

**Project administration:** Verena Braun.

**Resources:** Simon Hanslmayr.

**Software:** Mircea van der Plas, Benjamin Johannes Stauch.

**Supervision:** Simon Hanslmayr.

**Writing – original draft:** Mircea van der Plas.

**Writing – review & editing:** Benjamin Johannes Stauch, Simon Hanslmayr.

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
