## [Editor Report · Decision Letter 0]

11 Mar 2021

Dear Simon, 

Thank you for submitting your manuscript entitled "Slow rTMS to the left DLPFC enhances verbal memory formation" for consideration as a Research Article by PLOS Biology.

Your manuscript has now been evaluated by the PLOS Biology editorial staff, as well as by an academic editor with relevant expertise, and I am writing to let you know that we would like to send your submission out for external peer review. Please accept my apologies for the delay in sending this decision to you.

Please re-submit your manuscript within two working days, i.e. by Mar 15 2021 11:59PM.

Kind regards,

Gabriel Gasque, Ph.D.,

Senior Editor

PLOS Biology

---

## [Decision Letter · Decision Letter 1]

12 Apr 2021

Dear Simon,

Thank you very much for submitting your manuscript "Slow rTMS to the left DLPFC enhances verbal memory formation" for consideration as a Research Article at PLOS Biology. Your manuscript has been evaluated by the PLOS Biology editors, by an Academic Editor with relevant expertise, and by two independent reviewers. We will note that reviewer 1, Simon Davis, has revealed his identity.

In light of the Academic Editor's detailed comments and of the additional reviews (below), we will not be able to accept the current version of the manuscript, but we would welcome re-submission of a much-revised version that takes into account the Academic Editor and reviewers' comments. We cannot make any decision about publication until we have seen the revised manuscript and your response to the Academic Editor and reviewers' comments. Your revised manuscript is also likely to be sent for further evaluation by the reviewers.

We expect to receive your revised manuscript within 3 months. 

**IMPORTANT - SUBMITTING YOUR REVISION**

Your revisions should address the specific points made by the Academic Editor and by each reviewer. As you will see from their comments, the reviewers have noted that your findings are exciting, well-presented, and interesting. However, the reviewers have raised a number of concerns which will need to be addressed before we can consider your manuscript for publication at PLOS Biology. For example, reviewer 1 has raised a number of issues with the EEG and behavioral analyses which will need to be addressed to strengthen the conclusions of the study. Additionally, the reviewers identified a number of areas requiring additional clarification and discussion --particularly, the need of a more comprehensive description of the methods/procedures-- and raise concerns with a number of interpretations of the data. Please address these points thoroughly. 

Please submit the following files along with your revised manuscript:

*Re-submission Checklist*

*Published Peer Review*

*PLOS Data Policy*

*Blot and Gel Data Policy*

Sincerely,

Gabriel Gasque, Ph.D.,

Senior Editor,

ggasque@plos.org,

PLOS Biology

REVIEWS:

Reviewer #1: In their work "Slow rTMS to the left DLPFC enhances verbal memory formation," van der Plas, Braun and colleagues report an unexpected beneficial effect of noninvasive brain stimulation on memory formation. In an analysis of previously published work, the authors demonstrate that 1 Hz rTMS caused increased recall of items studied during concurrent stimulation of the DLPFC as opposed to the vertex. The beneficial effects of this stimulation were not observed for items studied in the previous list, suggesting a causal effect. Analysis of simultaneously collected scalp EEG data showed that stimulation influenced power decreases in the beta band at posterior sensors, an effect believed to reflect item processing. In a second experiment, the authors attempted to replicate the effect of 1 Hz rTMS on behavior, using a within subject design. Although there were deviations in the effects of stimulation on behavior (e.g., differences in memory performance were no longer limited to stimulation lists), DLPFC stimulation once again increased memory performance compared to the vertex control. My general impression is that this study describes an interesting behavioral effect of stimulation in a straightforward, clear manner. However, there are issues of analysis and interpretation of the EEG and behavioral data that make me question whether the claims argued by the authors are in fact supported by the data. With clarification of these issues, I believe this work would make a compelling case for beneficial effects of 1 Hz rTMS stimulation on memory.

Major Issues:

1) The major claim of the paper is that DLPFC stimulation affects encoding at the item level, potentially through feedback interactions with processing in posterior cortical regions. Behavioral evidence for this claim comes by comparing recall rates in Experiments 1 and 2, as well as measures of temporal organization in Experiment 2, to rule out an effect of stimulation on other memory processes. However, in the experimental paradigm recall of both list 1 and list 2 items occur at the same time. As a result, recall performance on the stimulated and non-stimulated lists are no longer independent from one another. This complicates the ability to interpret both the behavioral processes at play and the effect of stimulation. Given the delayed recall paradigm (and the resulting tendency to initiate recall at the beginning of list 1), recall of list 1 items can interfere with the ability to recall items on list 2. One potential mechanism by which stimulation may modulate recall performance is by reducing interference between the two lists. For example, DLPFC could elicit more similar contextual states across lists 1 and 2 in the DLPFC as compared to vertex stimulation condition. This contextual similarity would boost recall for list 2 in the DLPFC condition, while making it more difficult to access list 2 items in the vertex condition. This is just one of many potential behavioral explanations for the findings. In ruling out alternative explanations, I believe the authors only compared temporal clustering measures for list 2 between DLPFC and vertex conditions. To help interpret these results, it would be helpful to describe more accurately what was done, as well as report measures on each list (e.g., what were temporal clustering scores for each condition and list). In addition, it would be useful to compare clustering by list (i.e., did subjects cluster items from list 1 or list 2 together in their recall sequences) in each stimulation condition to rule out other potential contextual effects. In general, additional forms of evidence are necessary to make the claim that the effects are at a "word-specific" level. 

2) The stimulus pool consists of 240 nouns from the MRC psycholinguistic database. From my reading of the methods, it does not appear semantic similarity was controlled for on a list-by-list basis or between stimulation and non-stimulation lists for a single recall trial. Is it possible that any of the observed effects were related to semantic organization of recall?

3) I am not convinced that the reported EEG results, including the analysis of model parameters from the FOOOF toolbox, provide an accurate description of how stimulation affected neurophysiology. The authors are well-motivated in their analysis of the beta band (13-30 Hz) from prior work. However, from inspection of the power spectra reported in Figure 4b, there appear to be sideband effects from alpha oscillations that bleed into this frequency range. The reported effect of stimulation on memory may in fact be explained by this effect. Are effects in this beta band correlated with effects in the alpha range? Are there oscillatory effects of stimulation at alpha (i.e., does the amplitude of alpha oscillations from the FOOOF analysis differ between these conditions)?

4) The authors chose to make inferences on the power spectra based off parameters in the 1/f fit, finding a potentially interesting effect in the slope in log-log space. However, it is difficult to interpret these model parameters for several reasons. First, how well were the power spectra fit by the model? Differences between conditions could in fact be driven by issues with model fit rather than actual changes in the power spectra. A comparison of residuals between experimental conditions, broken down by frequency, would alleviate this concern. Second, is it possible to distinguish whether the effects of stimulation are during stimulus processing itself vs. the pre-stimulus period? As TMS was randomly timed relative to stimulus presentation, it would be useful to know whether the effects are related to stimulus processing, as interpreted by the authors.

5) For the EEG analysis, the authors use a cluster-based permutation test to identify sensors that show differences in beta power between the DLPFC and vertex groups. Given the between-subject design of the experiment, and the claim that decreases in beta power reflects word-specific processing, would the relevant neural measure not be differences in beta power for subsequently recalled and not recalled items? If beta desynchronization is similar for subsequently recalled and forgotten items in the DLPFC condition or there are no differences in subsequent memory effects across stimulation conditions, this would have strong implications for the mechanism by which stimulation affected memory. 

6) Regarding scholarship, the idea that 1 Hz rTMS is generally inhibitory has recently been challenged (see Castrillon et al., 2020, Science Advances), as the effects depend on the network structure of the targeted region. Notably, stimulation of frontal sites with network properties similar to the DLPFC location in the current study decreased local inhibition and disrupted network connectivity. Discussion of whether the present findings support inhibition as the mechanism by which 1 Hz stimulation affects brain function may be warranted.

7) The authors seem to imply that the effects of stimulation were specific to beta, as they state no effects were observed for alpha or theta frequencies. However, I assume this is because the authors filtered their analysis to sensors that showed a beta difference in the 1 second post item onset. If the authors use the same selection procedure for alpha or theta bands, are there no observed effects?

Minor issues:

1) It would be helpful to clarify the prediction justifying a one-tailed test for DLPFC vs. vertex effects in the methods.

2) It is unclear if the FOOOF analysis included frequencies above 40 Hz or below 2 Hz during model fitting. The methods claim 2-35 Hz, but the figures do not match this description. If these frequencies were excluded, they should be excluded from Figures 4B-D as the model was not fit to these frequencies.

3) It is unclear what the R graphic denotes when reading Figure 1. It would be helpful for the reader if this were explained in the caption.

4) How were EEG data z-scored (e.g., across trials or time, within condition or across conditions, across all items or only recalled items)? The translation from the relatively raw data in Figure 4b to the time-frequency plot in Figure 3a are surprising, given the lack of any alpha signals in the latter.

Reviewer #2 Simon Davis: The current revision describes a novel analysis of two sets of EEG-TMS data collected while healthy adults (N = 20, 24) performed an episodic encoding (list-learning) task, in the context of a larger directed forgetting paradigm. Online TMS pulses were delivered at 1 Hz to the left DLPFC throughout stimulus presentation, which resulted in superior subsequent recall, compared to controls who received the same stimulation at vertex; an additional follow-up replicated the effect with a within-subjects design. The strength of this manuscript rests on not only the improved memory function, but also novel findings describing the role of beta power modulation in response to TMS. Somewhat orthogonal to the authors' well-known sync-desync model, these effects were instead a result of a spectral tilt, i.e., the aperiodic component of the power spectrum. These results therefore offer a new perspective on the role of frequency in memory function, and the manuscript is a great example of

re-examining older data with a new analytical lens. Furthermore, the presentation of these data is excellent, many different kinds of behavioral and complex EEG-related findings are presented clearly and intuitively; the explanation of spectral tilt is especially useful. I have only minor suggestions.

Though it is acknowledged throughout the Intro/Disc/Methods that the original experiment was a directed forgetting paradigm, it would be useful to mention when the paradigm is first explained (e.g. near ln77) that the data analyzed here represents ½ of the previous data set, i.e. the runs without the "Forget" cues.

As stated in the original manuscript, "there was no systematic temporal coupling between the delivery of the TMS pulses and the List 2 items." Was the presence of TMS artifact therefore even over the course of the epoch? This seems to be the assumption.

More detail on how "data around the rTMS artifacts [was] removed" (ln416) would be helpful. E.g., how did the authors remove these high-amplitude signals? Were they replaced with noise/zeros? How much time (in ms) does "around" mean in the above sentence?

ln169: "Figure 5" should be "Figure 4"

ln274, 287; The mechanistic inference between spectral tilt and parietal inhibition is ambiguous; the authors could clarify that the spectral slope is assumed to reflect the ratio between excitation and inhibition at the synaptic level with more negative slopes reflecting enhanced inhibition (Miller, PLoS Comp Biol, 2009).

Signed,

Simon Davis

Duke University

Academic Editor: In this manuscript, the authors report testing the effects of 1 Hz rTMS to the left DLPFC in two simultaneous EEG-TMS experiments, where the second study served as a replication of an incidental finding in the first study. Across both studies - and further confirmed meta-analytically on the combined data -, the authors found enhanced memory for word lists in the DLPFC stimulation compared to control stimulation of the vertex. They further report this effect to be associated with stronger beta power modulation in posterior regions, likely stemming from spectral tilt rather than from oscillatory mechanisms. 

Overall, the study is timely, reports exciting results, and is methodologically sound. In particular the strategy to back up an incidental finding by a replication study increases confidence in the results. I see a number of points - in particular in the presentation of methods and in the discussion - that need some attention before publication though.

1. General overview: The experimental design and overall protocol of the study is very difficult to grasp without going back and forth between introduction, results and methods sections. To aid the reader, I would suggest to give more information already earlier in the manuscript, and present the design and overall protocol in a more detailed schematic than currently done with Figure 1. For example, include information about the number of runs in the results already, as the reader in the current presentation might get the impression that the experiments included only 2x10 words in total.

2. Timing of effects: I had some problems grasping the exact timing of the EEG analyses, both regarding methods and interpretation. In particular, to which time windows were the statistical analyses restricted - just the 1s post-stimulus (as indicated on p.23), or also before (as indicated in figure 3)? In the discussion, the authors interpret EEG differences after stimulus, however not those preceding the stimulus, which however seem in stronger need for an explanation. In the methods description, the authors report that there was no relationship between the timing of the rTMS pulses and the stimulus presentation, however do they have any timing information available, e.g. pulse triggers, which might help to shed some light on the timing of effects? 

3. General interpretation of effects: Given that brains tend to be optimized during evolution, enhancement effects through brain stimulation are more interesting, but also more difficult to explain than impairing effects. If I understood it correctly, the authors hypothesize that their rTMS protocol led to disinhibiting effects. However, again from an evolutionary point of view, decreasing an existing inhibition should be even easier to evolve than a novel positively exciting effect - which makes the results somewhat puzzling. I would find it helpful if the authors could elaborate a bit more on two points in this regard: 

a) in how far can the authors exclude the possibility that vertex stimulation decreases rather than DLPFC stimulation increases memory performance? 

b) Which memory-relevant processes specifically might be (dis-)inhibited, and in how far would this lead to e.g. more efficient neural coding?

4. Blinding: The authors present the second study - already in the abstract - as conducted double blind. However, they use a rather non-standard definition of double blinding, namely simply being blind towards the specific hypotheses. According to this definition, likely the majority of studies would have to be considered at least single-blind, and many studies where data acquisition is done by research assistants with little involvement in the design and interpretation would have to be considered double blind this way. Under this definition, even participants of a pharmacological study who receive open label medication would have to be considered blinded as long as they aren't told explicitly which effects have been hypothesized for verum vs. placebo - which certainly would be anything but double blind according to the common use of the term: namely that participants and researchers can't tell which condition they are in during the experiment. I would thus recommend not to speak of 'double blind' altogether, and/or at least make it very clear throughout the abstract and manuscript how this term is used here. 

Minor: 

In Figure 4A, the y axis lacks a label.

---

## [Decision Letter · Decision Letter 2]

24 Jun 2021

Dear Simon,

Thank you for submitting your revised Research Article entitled "Slow rTMS to the left DLPFC enhances verbal memory formation" for publication in PLOS Biology. I have now obtained advice from the original reviewers and have discussed their comments with the Academic Editor. 

Based on the reviews, we will probably accept this manuscript for publication, provided you satisfactorily address the remaining points raised by reviewer 1. Please also make sure to address the data and other policy-related requests listed below my signature.

We expect to receive your revised manuscript within two weeks. 

*Published Peer Review History*

*Early Version*

Sincerely,

Gabriel Gasque, Ph.D.,

Senior Editor,

ggasque@plos.org,

PLOS Biology

TITLE: We think that for our broad readership a decompressed title will be more appealing. Thus, we suggest: 

Stimulation of the left dorsolateral prefrontal cortex with slow rTMS enhances verbal memory formation

FINANCIAL STATEMENT

Please include in the submission system the funders’ websites.

BLURB

Please provide one.

ETHICS STATEMENT:

>> Please include within your manuscript the ID number for your protocol approved by the ethics committee of the University of Konstanz.

>> Please indicate if the form of consent given by the participants was written or oral. If it wasn’t written, please explain why.

>> Please indicate if your experimental protocol adhered to the principles expressed in the Declaration of Helsinki or to any other specific national or international ethical guidelines.

DATA POLICY:

Note that we do not require all raw data. Rather, we ask for all individual quantitative observations that underlie the data summarized in the figures and results of your paper. For an example see here: http://www.plosbiology.org/article/info%3Adoi%2F10.1371%2Fjournal.pbio.1001908#s5

These data can be made available in one of the following forms:

Regardless of the method selected, please ensure that you provide the individual numerical values that underlie the summary data displayed in the following figure panels: Figures 2A-E, 3A-D, 4B-D, 5A-D, 6AB, S1, S2A-D and S3.

Please also ensure that each figure legend in your manuscript includes information on where the underlying data can be found and that your supplemental data file/s has/have a legend.

DATA NOT SHOWN?

Reviewer remarks:

Reviewer #1: The authors have done an impressive job responding to reviewer concerns and have greatly improved the manuscript. I only have two minor comments that are non-essential in nature:

1. The additional detail regarding the temporal clustering analysis is helpful for the reader. However, it was still unclear to me how temporal distances were computed. Specifically, it is unclear whether 'position during encoding' and expected distance resets at the beginning of each list (i.e., does the first item in List 2 start at position 1 or position 11). This information is necessary to interpret the temporal clustering results. This detail is not provided in the referenced paper, which does not have multiple lists per recall period. Thus, it is unclear whether this measure provides information regarding clustering between as opposed to within the two lists. 

2. In the discussion, the authors mention (line 345) that the memory effect (of stimulation) was specific for list 2 words. However, the behavioral analysis showed a positive effect of stimulation on items at the end of list 1 in experiment 1. Although the experimental design makes it difficult to interpret this effect (because both lists were recalled at the same time, see my previous major concern 1), this point seems rather important in interpreting the effects of stimulation, and I would tone down this claim. However, these are exciting results, and I am curious to know how and whether they replicate outside of the directed forgetting paradigm.

Reviewer #2, Simon W Davis: The additional clarifications on the task design, TMS timing, and on the removal of TMS artefacts have greatly improved the manuscript. The authors have addressed all my extant concerns.

---

## [Editor Report · Decision Letter 3]

14 Jul 2021

Dear Simon,

On behalf of my colleagues and the Academic Editor, Martin Dresler, I am pleased to say that we can in principle offer to publish your Research Article "Stimulation of the left dorsolateral prefrontal cortex with slow rTMS enhances verbal memory formation" in PLOS Biology, provided you address any remaining formatting and reporting issues. These will be detailed in an email that will follow this letter and that you will usually receive within 2-3 business days, during which time no action is required from you. Please note that we will not be able to formally accept your manuscript and schedule it for publication until you have made the required changes.

***IMPORTANT:

1) I updated your submission in the Editorial Manager to include the final title and the OSF link (https://osf.io/dyxjv/). Please check that you are satisfied with these changes.

2) Please ensure that in your final version of the manuscript, each figure legend includes information on where the underlying data can be found: https://osf.io/dyxjv/

PRESS

Sincerely, 

Gabriel Gasque, Ph.D. 

Senior Editor 

PLOS Biology

ggasque@plos.org